# The Influence of a Psychosocial Rehabilitation Program in a Community Health Setting for Patients with Chronic Mental Disorders

**DOI:** 10.3390/ijerph18084319

**Published:** 2021-04-19

**Authors:** Paweł Rasmus, Anna Lipert, Krzysztof Pękala, Małgorzata Timler, Elżbieta Kozłowska, Katarzyna Robaczyńska, Tomasz Sobów, Remigiusz Kozłowski, Michał Marczak, Dariusz Timler

**Affiliations:** 1Department of Medical Psychology, Medical University of Lodz, 90-419 Lodz, Poland; pawel.rasmus@umed.lodz.pl (P.R.); krzysztof.pekala@umed.lodz.pl (K.P.); tomaszsobow@yahoo.com (T.S.); 2Department of Sports Medicine, Medical University of Lodz, 90-419 Lodz, Poland; anna.lipert@umed.lodz.pl; 3Department of Management and Logistics in Healthcare, Medical University of Lodz, 90-419 Lodz, Poland; malgorzata.timler@stud.umed.lodz.pl (M.T.); michal.marczak@umed.lodz.pl (M.M.); 4Department of Experimental Immunology, Medical University of Lodz, 90-419 Lodz, Poland; elzbieta.kozlowska@umed.lodz.pl; 5Department of Health Care Management, Medical University of Lodz, 90-419 Lodz, Poland; katarzyna.robaczynska@umed.lodz.pl; 6Center of Security Technologies in Logistics, Faculty of Management, University of Lodz, 90-136 Lodz, Poland; remigiusz.kozlowski@wz.uni.lodz.pl; 7Department of Emergency Medicine and Disaster Medicine, Medical University of Lodz, 90-419 Lodz, Poland

**Keywords:** mental health, rehabilitation, care, health services research

## Abstract

*Purpose*: To examine (a) the amount of health-related behavior, (b) the level of generalized optimism, (c) the belief about patients’ abilities to cope with difficult situations and obstacles and (d) the subjective sense of social exclusion at baseline and at follow-up among patients with chronic mental health issues participating in a psychosocial rehabilitation program in a community mental health setting. *Materials and Methods*: This prospective study involved 52 participants aged 18–43 years and diagnosed with mental illness who participated in a 6-month psychosocial rehabilitation program, organized within a special community setting. Different questionnaires were used: the Health-Related Behavior Questionnaire, the Revised Life Orientation Test, the General Self-Efficacy Scale, the Personal Competence Scale and a self-made questionnaire concerning social exclusion problems. *Results*: Statistical analysis of the questionnaire results taken at the beginning and end of the six-month course, running from November 2015 to May 2016, revealed significant increases in health-related behavior (*p* = 0.006) and general self-efficacy (*p* = 0.01). *Conclusions:* Psychosocial rehabilitation programs offered by community mental health settings might serve as an easy, accessible strategy to deal with different interpersonal and intrapersonal problems and as a potential way to improve health behavior. Further research is required to evaluate other psychosocial rehabilitation programs in different community mental health settings in Lodz Voivodeship, Poland.

## 1. Introduction

Approximately 14% of the global burden of disease has been attributed to neuropsychiatric disorders [1]. For example, serious mental illness (SMI) affects 1–2% of the population [2], and people with SMI have an average lifespan that is 12.5 years shorter than that of the general population [3]. Mental illnesses are seen as a considerable problem for a patient, mostly due to their negative influence on daily functioning, performance of social roles and quality of life [4]. As the course of illness is quite often intense and hard to control, chronically mentally ill patients can often experience social exclusion [5,6].

A patient with chronic mental illness has different cognitive and functional inabilities that cause behavioral problems and make it impossible to fulfill roles and responsibilities [7]. Although the majority of chronically mentally ill patients have a diagnosis of schizophrenic disorders, other patient groups with psychotic and non-psychotic disorders [8,9,10] are targeted by psychiatric rehabilitation [11]. A major challenge in psychiatric care is to create an open and rehabilitative environment that promotes patient recovery [12] and that involves redefinition of one’s illness [13]. These people need to access services that are not only effective in treating their mental health but also increase their awareness of lifestyle choices and promote autonomy and independence, thereby reducing their need for inpatient services. Therefore, it is important to develop appropriate procedures of psychosocial rehabilitation that put an emphasis on improving patient functioning in various spheres of life [6,14,15] Rehabilitation interventions concern the so-called “subjective” model of recovery and, thus, promote taking an active position against the illness, which encourages self-determination and empowerment [16]. In psychosocial rehabilitation, a lot of tools can be used—for example, case management, supported employment, cognitive remediation, psychoeducation and cognitive behavioral therapies [13].

Community mental health facilities have been created for people whose mental or intellectual disability limits their everyday life, education, work or performance of social roles [17]. The facilities provide both everyday and special care for their clients, who can be classified as follows: Type A—chronically ill people; Type B—people with mental retardation; Type C—people with other chronic mental problems. These institutions provide special psychosocial rehabilitation methods adjusted to the needs of the participants to build resourcefulness, increase self-reliance and to integrate clients with society [18]. Various forms of training are available, addressing areas such as daily life routines that build self-reliance and independence, social skills, interpersonal skills and how to spend free time—for example, by expanding interests, reading magazines, etc. Many other forms of therapy are available to prepare the participant for occupational therapy or employment, such as physical activity, help with official matters and access to health benefits, counseling and training in financial skills [13,19,20].

It is extremely important to understand the role of psychosocial rehabilitation programs. Those are treatment approaches “designed to help improve the lives of people with disabilities”. Their goal is to develop emotional, cognitive and social skills to make it easier for people diagnosed with mental illness to thrive in their communities [21]. One of the basic ideas is to minimize their sense of exclusion and help them become a real part of those societies. Studies show that there are many ways to check the efficacy of psychosocial rehabilitation programs by verifying factors such as improvement of residual symptomatology, remediation of cognition and social skills, cognitive remediation, support for occupational integration and improvement of everyday life activities [21]. As qualities such as optimism and development of coping mechanisms seem to be important factors for progress when working toward each of those goals, it seems rational to examine whether a program is working toward also achieving those goals.

Although rehabilitation of mentally ill patients is an essential component of psychiatric management, the effectiveness of such programs has to be further explored. Therefore, the aim of this study was to examine (a) the amount of health-related behavior, (b) the level of generalized optimism, (c) the belief about patients’ abilities to cope with difficult situations and obstacles and (d) the subjective sense of social exclusion at baseline and at follow-up among patients with chronic mental issues participating in a psychosocial rehabilitation program in a community mental health setting.

## 2. Methods

### 2.1. Sample

In total, 52 participants aged 18–43 years and diagnosed with mental illnesses qualified to participate this prospective study. All of the individuals were the residents of one of the biggest voivodeships in Poland. The inclusion criteria comprised the following: (1) diagnosis of a chronic mental illness; (2) ability to read and understand questions with the support of an assistant if needed; (3) acceptance to participate with at least 80% presence during a 6-month psychosocial program provided by a community mental health institution. The exclusion criteria included the following: (1) lack of any diagnosis of mental illness; (2) greater than 20% absence during psychosocial program activities, controlled by an attendance list. Finally, the data obtained from 22 participants (8 women and 14 men) with a mean age of 28 ± 6.59 years (30 ± 7.13 and 26 ± 4.83 years, respectively) were analyzed. Thirty participants did not finish the whole 6-month program because of serious health problems, hospitalization or their employment preventing further participation in the program. Written informed consent was obtained after the procedures had been fully explained. Participants were informed about the voluntary and anonymous nature of the study. No financial incentives were awarded for taking part in the study. The study was approved by the Institutional Review Board of the Medical University of Lodz before the beginning of the assessments, approval Ref: (no. RNN/292/15/KB).

### 2.2. Procedure

The psychosocial program was organized by a specially designed community mental health setting created in 2006 in Lodz, being one of only a few of this kind in the country. It is a Type A institution whose goal is to improve the everyday functioning of people with mental disorders and assist their integration with society. The facility includes 26 services. It employs psychologists, pedagogues, occupational therapists, physiotherapists and other specialists according to the needs of the center. The participants are recruited on the basis of a decision by the Municipal Social Services Center, after meeting with the head of the community mental health facility. They are referred by their psychiatrist (most of the time) and they have a certificate from an internist about the lack of contraindications to participation in the activities organized by the institution. Although some participants have a lower intelligence quotient (IQ), most are of an average IQ without any substance addiction or have undertaken a long period of abstinence. This community mental health setting offers both group training and individual meetings. They are organized from Monday to Friday from 10 a.m. to 4 p.m. Every day is planned to include one group training session and one individual meeting. The range of services available for participants includes training of daily life routines (taking care of appearance, hygiene training, cooking training and training of financial skills), training for interpersonal skills and problem solving, training on how to spend free time, counseling, help with official matters, help with gaining access to health benefits, training for caregivers and physical activity, as well as other forms of therapy. This association also prepares young people with mental problems for the best possible life in society. The program helps in building self-esteem and relationships with other people. This is one of the standard forms of environmental treatment. The community mental health setting provides the possibility to participate actively in organizing various events, for example, Christmas or Easter meetings. The institution also offers the possibility to take part in competitions, festivals or exhibitions organized by various organizations. These activities are intended to help participants learn how to cope with shyness, stress or closeness and allow the possibility to express both positive and negative emotions.

The participants of the study were obliged to take part in all activities that were organized in the facility during 6 months. A contract to this effect was signed during meetings with the caregiver/supervisor. Changes in individual contracts were only possible due to health reasons or when a participant was engaged in certain forms of activity, such as participation in other courses.

The study was longitudinal, with the first part conducted from November 2015 to May 2016. The research was carried out using a diagnostic poll method with four different questionnaires: the Health-Related Behavior Questionnaire, the Personal Competence Scale, the Revised Life Orientation Test, the General Self-Efficacy Scale and a self-made questionnaire concerning social exclusion problems. The questionnaires were completed by the participants at the beginning and end of the 6-month period of the rehabilitation program on site at the Community Health Setting.

The Health-Related Behavior (HRB) Questionnaire [22,23,24] is a questionnaire that consists of 24 statements describing different behaviors related to health. These statements are divided into four categories: eating habits, preventive behavior, health practices and positive mental attitude. All categories are measured by the self-reported frequency of certain activities. The respondent marks the frequency of performing each activity during the previous year on a five-point scale (1—almost never; 2—seldom; 3—from time to time; 4—often; 5—almost every day). The final score ranges from 24 to 120, with higher scores being associated with more intense health-related behavior.

The Personal Competence Scale (PCS) [22] was designed to assess general self-efficacy as well as its components, such as a feeling of strength to start an action (subscale A) and prevailing to continue it (subscale B). Each subscale includes 6 statements of both positive and negative meanings. The sum of the points referring to each of the responses was calculated (from 1 to 4 points, depending on how the response was formed). This made it possible to estimate a general outcome of a feeling of self-efficacy (12–48 points) as well as the parameters of feeling strong and persevering to complete the task (for each of 12–48 points). The raw scores of the test were converted to sten scores of 1–10 (a standard scoring system). Interpretation of the sten scores is as follows: 1–4 represent low, 5–6 represent medium and 7–10 represent high test results. To assess the feelings of strength and perseverance, higher scores represent a greater feeling of strength/perseverance, and on the other hand, lower scores refer to a lesser feeling of strength/perseverance [25].

The Revised Life Orientation Test (LOT- R) [26,27,28,29] comprises 10 statements, six of which have a diagnostic value for generalized optimism. The LOT-R is designed to study all adults, irrespective of their health status. The respondent marks their agreement with the statements on a five-point scale: A = I strongly agree; B = I agree a little; C = I neither agree nor disagree; D = I disagree a little; E = I disagree a lot. The result of the questionnaire is a description of the level of generalized optimism, with the raw scores set between 0 and 24 points. Again, a higher score indicates a higher level of optimism. The questionnaire has an acceptable level of internal consistency (Cronbach’s alpha 0.78) [30].

The General Self-Efficacy Scale (GSES) [28,31] is intended to provide a general opinion of the efficacy with which the respondent deals with difficult situations and obstacles. The GSES is designed for adults. It contains 10 statements, answered on a four-point scale from “NO” (1 point) to “YES” (4 points), giving a final score between 10 and 40. Again, a higher score indicates a higher sense of self-efficacy. This questionnaire is correlated to emotion, work, optimism and satisfaction. Negative coefficients were also found for depression, health complaints, burnout, stress and anxiety. The questionnaire also has an acceptable level of internal consistency (Cronbach’s alpha 0.76–0.90) [31].

A self-made questionnaire concerning social exclusion problems was designed. It consisted of eight yes/no questions concerning the social and familial functioning, financial status and civic and cultural life of the respondent. The final score acted as a generalized index of the sense of social exclusion, ranging from 8 to 16 points. Higher scores were associated with a greater sense of social exclusion.

All of these tools are self-rated pen-and-paper questionnaires. All four were chosen for this particular group of subjects, taking into consideration their cognitive and incentive functioning. Both the accuracy and reliability of these tools are sufficient for this kind of study. These tools were also used because of their simplicity.

### 2.3. Data Analysis

Normality of distribution was tested with the Shapiro–Wilk test. Considering the lack of normal distribution and the small size of the group, non-parametric tests were used [32]. The Wilcoxon test [32] was used to determine the statistical significance of the results from all questionnaires. Linear regression analysis could not be performed because of the distribution of the studied dimensions and the sample size; therefore, only the Spearman correlation coefficient was used to assess the dependence between the studied variables. The effect size measure of differences between the results from the beginning and end of the 6-month rehabilitation was verified using Cohen’s d test. It is defined as the difference between two means divided by a standard deviation for the data. Cohen classified effect sizes as small (d = 0.2), medium (d = 0.5) and large (d ≥ 0.8). This means that if two groups’ means do not differ by 0.2 standard deviations or more, the difference is trivial, even if it is statistically significant [33]. All statistical analyses were performed using STATISTICA 12.0 PL, and *p* < 0.05 was considered as statistically significant.

## 3. Results

The study group constituted 22 participants, including 14 (fraction 0.64) men and 8 (fraction 0.36) women, with an average age of 28.36 (6.59) years (29.86 (7.13) and 25.75 (4.83) years, respectively).

The results of the HRB questionnaire obtained in November and in May indicated a statistically significant improvement in different health behaviors (*p* < 0.01), such as preventing behavior (*p* < 0.05), positive mental attitude (*p* < 0.001) and health practices (*p* < 0.0) (Table 1). There was a positive change in eating habits, but it was not statistically significant (Table 1). The results from the GSES questionnaire showed a significant improvement in self-efficacy (*p* < 0.05) (Table 1). An improvement in general optimism and sense of social exclusion was noticed, but the differences in the results obtained in November and in May were not statistically significant (Table 1).

The sense of self-efficacy, perseverance and general optimism were noticed to be improved after the 6-month rehabilitation program (Figure 1). Additionally, a positive change in the health-related behaviors was also noticed (Figure 1).

While analyzing the results obtained in November 2015 and in May 2016, most of the correlations between the variables seemed to increase. In particular, it was noticed in relation to perseverance and feeling of strength, optimism, sense of social exclusion and self-efficacy (Table 2 and Table 3).

The sense of social exclusion was significantly lower in people with a greater sense of self-efficacy. There was also a negative correlation between optimism and social exclusion, and the obtained results were statistically significant both in November and in May. A very strong positive correlation was observed between health behaviors and the level of optimism and sense of self-efficacy. A sense of social exclusion could also negatively influence the health behaviours of the study participants. In addition, a statistically significant positive correlation between perseverance and feeling of strength was observed (Table 2 and Table 3).

## 4. Discussion

This study mainly concerns the influence of a psychosocial program specially designed in the community mental health setting, whose task is to improve social skills, increase motivation in terms of health behavior and increase sense of self-efficacy. To our knowledge, there are no studies investigating the efficacy of this kind of program organized in the way described in the present study. Therefore, findings from this study will contribute to the existing research.

During participation in the psychosocial rehabilitation program, the greatest emphasis was put on maintaining a healthy weight and limiting the use of cigarettes and other psychoactive substances. It is estimated that frequency of smoking increases with a greater number of mental illnesses, ranging from 18% for people with no mental illness to 61% for people diagnosed with three or more mental illnesses [34]. In turn, taking part in physical activity (PA) classes not only contributed to the subject acquiring a better frame of mind and healthy lifestyle habits but also enabled stress release. It was confirmed that people who followed vigorous PA recommendations were less likely to report poor mental health and perceived stress [35]. For example, in a study among psychiatric patients meeting the criteria for major depression, an 8-week hatha yoga intervention resulted in statistically and clinically significant reductions in depression severity [36]. Other improvements in different abilities were also observed. Leading a healthy lifestyle through recreational sports not only allowed the participants to establish new relationships, but it also fostered the development of social ties and the need to exchange opinions. The participants were also engaged in cooking classes, during which they were given information about the rules of healthy eating, the caloric content of meals and the importance of a varied diet. They also acquired knowledge of safety and hygiene while preparing meals.

Various determinants of health need to be taken into consideration to evaluate healthy behavior in a group of people with mental disorders. It is confirmed that the main motivation for undertaking healthy behavior, especially preventive behavior, is to maintain good health behavior [37]. Some studies have indicated that some types of health-related behavior are undertaken for other reasons—for example, dieting or physical exercise might be performed to improve changes in appearance resulting from the side effects of some pharmaceuticals [14].

Our findings indicated that participation in the psychosocial rehabilitation program affected the generalized optimism experienced by participants of the services of the community mental health setting. The results of the present study suggest that it might be difficult, but not impossible, to modify generalized optimism via a psychosocial rehabilitation program. It is important, because the level of optimism plays a modifying role in a range of activities and improves health outcomes of chronically ill people [38]. It mostly correlates positively with high self-esteem, self-worth and self-efficacy and internal locus of control but negatively correlates with depression, helplessness and anxiety [18]. In a study using a similar tool (LOT-R questionnaire), but in the psychiatric care setting, it was also noticed that lower levels of dispositional optimism are associated with stage of affective disorders, even after remission [39].

Our findings showed that the studied rehabilitation program significantly influenced the ability of the participants to cope with difficult situations and obstacles in a positive way, including those affecting health-related behavior. The participants were found to have significantly greater faith in their own abilities by the end of the course, which favors achieving goals and releasing life energy. Many previous studies confirmed that self-efficacy influences intentions and actions regarding various health-related behaviors [40,41,42], which is especially important in people with mental illnesses.

Mentally ill people frequently face social exclusion [43] and, as a consequence, report feeling lonely and rejected. Moreover, it was noted that more severe cases of mental disorders are associated with worse social functioning [44]. All of this has significant consequences on the effects of psychosocial rehabilitation programs, associated with the compliance, adherence and eventual recovery and quality of life of the participant [10]. Other studies indicate that psychosocial rehabilitation programs are particularly strongly influenced by psychoeducation and interventions concerning preventive and therapeutic social support systems [45,46]. That is why it is noteworthy that the results of the present study demonstrated the presence of changes in social exclusion.

The main strength of this study was the uniqueness of the procedure itself, i.e., the identification of changes in social skills, motivation in terms of health behavior and sense of self-esteem among people undergoing the specially designed psychosocial program during the 6-month period. Moreover, the study was performed using four different international questionnaires.

However, some marginal notes have to be made. First, the sample size was small, which can influence the methods and obtained results. That is why the effect size value was calculated to evaluate the statistical significance. If the effect size of an intervention is large, it is possible to detect such an effect with smaller sample numbers. Secondly, there was a large age discrepancy in the study group, meaning that the participants might have been exposed to different social settings that could affect their health-related behavior and self-efficacy. Nevertheless, it should be pointed out that it was very difficult to collect a homogenous group of participants because of a phenomenon of high turnover of users in the community mental health setting. Next, it needs to be stressed that the durability of the effects obtained in the study cannot be predicted, as some forms of training, for example, cognitive process training, need to be repeated regularly to be effective. Thus, some abilities may regress if the program is discontinued. Another limitation is that we did not collect secondary information on other characteristics of the participants, i.e., pharmacotherapy used by the patient or other psychotherapeutic methods except those in the health center. The study had no control group. Therefore, further studies including a comparison group, which should consist of people excluded from rehabilitation, are needed. It should be emphasized that the purpose of this study was not to generalize the results obtained from the selected sample to the general population, but rather to understand a complex problem and use the results in similar or related situations. It is recommended to perform follow-up studies to determine the long-term efficacy of psychosocial rehabilitation programs. In this kind of study, the participants should be examined before, during and at the end of the psychosocial rehabilitation program and, additionally, 6 months after the end of the program.

There is a need for a gradual move from the biomedical psychiatry model to the biopsychosocial model of holistic rehabilitation of patients with mental disorders. The biopsychosocial model considers as a success not only a reduction in the number of symptoms or the number of hospitalizations, but mainly an improvement in the quality of life seen by patients and building of their higher self-reliance and resourcefulness as well in their personal and professional life [47]. Our findings suggest that there may be hope for the evolution of community psychiatry as an indivisible portion of professionally organized mental healthcare in a country. Further research is required to evaluate other psychosocial rehabilitation programs in different community mental health settings and to look at similar programs in different regions to determine the cultural effectiveness.

## 5. Conclusions

Psychosocial rehabilitation programs offered by community mental health settings might serve as an easy, accessible strategy to deal with different interpersonal and intrapersonal problems and as a potential effective way to improve health behavior, especially concerning preventive behavior, positive mental attitude, health practices and feelings of self-efficiency for psychiatric patients. Several key conclusions emerged from the present study. The psychosocial rehabilitation program significantly improved various health behaviors, such as behavior prevention, positive mental attitude and health practices. The level of generalized optimism also improved after the 6 months of rehabilitation. The psychosocial rehabilitation significantly contributed to the improvement in patient beliefs about their ability to cope with difficult situations and obstacles; at the same time, this program did not reduce the subjective sense of social feeling of exclusion among the patients participating in the study. As the data are still limited in that area, the qualified staff who work within psychosocial rehabilitation services should be encouraged and supported to undertake research. Only through developing an evidence base can those services demonstrate their efficacy. Although the results of the current study show some promise, a larger and comparative study, preferably a randomized control trial, is needed to address the question of effectiveness of the rehabilitation program.

## Figures and Tables

**Figure 1 ijerph-18-04319-f001:**
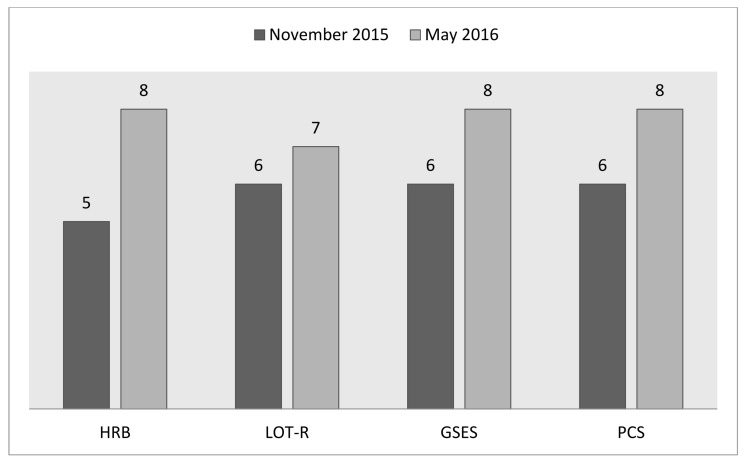
The results from four questionnaires at baseline and at follow-up among patients with chronic mental illness participating in a 6-month psychosocial rehabilitation program, presented as sten values.

**Table 1 ijerph-18-04319-t001:** Mean scores obtained from four questionnaires used at the beginning and end of the 6-month rehabilitation program.

Measured Features	Year 2015 before Rehabilitation (n = 22)	Year 2016 after Rehabilitation (n = 22)	*p*-Value	*d*	IC 95%
	Mean (SD)	Mean (SD)			
Health-related behavior	76.5 (12.83)	94.5 (12.07)	0.006	1.018	0.39–1.65
* Eating habits	18.5 (4.09)	22.5 (4.55)	0.055	0.710	0.10–1.32
* Preventive behavior	19.0 (10.65)	24.0 (4.40)	0.027	0.341	−0.26–0.94
* Positive mental attitude	19.5 (3.27)	25.0 (4.03)	0.001	1.220	0.58–1.86
* Health practices	19.5 (3.70)	24.0 (3.72)	0.002	0.882	0.26–1.50
Generalized optimism (LOT-R)	16.5 (4.42)	17.5 (4.36)	0.870	0.164	−0.43–0.76
Generalized self-efficacy (GSES)	29.0 (5.78)	33.0 (7.45)	0.010	0.502	−0.10–1.10
Social exclusion (self-made tool)	11.0 (0.91)	10.0 (2.15)	0.072	−0.797	−1.41–−0.18

Spearman’s correlation coefficient was used to assess the dependence between the studied variables. * Statistical significance was determined using Wilcoxon’s test when *p* < 0.05. *d* means Cohen’s d as effect size. Abbreviations: LOT-R, Life Orientation Test-Revised; GSES, General Self-Efficacy Scale.

**Table 2 ijerph-18-04319-t002:** Correlations between the studied variables in November 2015.

	HRB	PCS	PCS	PCS	LOT-R	GSES	Self-Made Questionnaire
Health behaviors (HRB)	-						
Self- effectiveness (PCS)	0.34	-					
Perseverance (PCS)	0.37	0.73 **	-				
Feeling of strength (PCS)	0.32	0.89 **		-			
Optimism (LOT-R)	0.60 **	0.28	0.13	0.28	-		
Sense of self-efficacy (GSES)	0.56 **	0.22	0.24	0.20	0.67 **	-	
Social exclusion (Self-made questionnaire)	−0.43 **	0.04	- 0.01	−0.04	−0.52 *	−0.65 **	-

Spearman correlation’s coefficient was used to assess the dependence between the studied variables. Statistical significance, determined by the Wilcoxon test, was analyzed. * statistical significance *p* < 0.05; ** statistical significance *p* < 0.01.

**Table 3 ijerph-18-04319-t003:** Correlations between the studied variables in May 2016.

	Health Behaviors (HRB)	Self-Effectiveness (PCS)	Perseverance (PCS)	Feeling of Strength (PCS)	Optimism (LOT-R)	Sense of Self-Efficac (GSES)	Social Exclusion (Self-Made Questionnaire)
Health behaviors (HRB)	-						
Self- effectiveness (PCS)	0.06	-					
Perseverance (PCS)	−0.02	0.73 **	-				
Feeling of strength (PCS)	0.10	0.89 **	0.40 *	-			
Optimism (LOT-R)	0.42 *	0.17	−0.18	0.28	-		
Sense of self-efficacy (GSES)	0.51 **	0.83 **	−0.31	0.64 **	0.56 **	-	
Social exclusion(Self-made questionnaire)	−0.55 **	0.04	0.25	−0.30	−0.67 **	−0.59 **	-

Spearman’s correlation coefficient was used to assess the dependence between the studied variables. Statistical significance, determined using the Wilcoxon test, was analyzed. * statistical significance *p* < 0.05; ** statistical significance *p* < 0.01.

## Data Availability

The data presented in this study are available on request from the corresponding author.

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
