# Peer review of "The Influence of a Psychosocial Rehabilitation Program in a Community Health Setting for Patients with Chronic Mental Disorders"

_ijerph, 2021, doi:10.3390/ijerph18084319_

Round 1

Reviewer 1 Report

There wasn't sufficient rationale on why it was important to examine optimism, coping mechanism, and sense of exclusion. Add another paragraph or two to discussed the lack of research in this area or the importance of these qualities in psychosocial programs.

Reviewer 2 Report

Title:  The Influence of a Psycho-Social Rehabilitation Program in a Community Mental Health Setting for Patients with Chronic Mental Health Issues.

Overall:  As mentioned in the study, there is plenty of evidence to support individualized approaches to therapy for individuals with chronic disorders, but few take all of them into account and include the influence of social exclusion. This is important and novel, especially with the emphasis on the holistic approach.  There needs to be more discussion about the limitations of the actual study. 

Abstract:  Clear summary of methods, results and general conclusions.

Introduction:

Overall:  A good overview of psycho-social programs available at most mental health facilities for individuals with chronic mental health disorders.  It would be helpful to clarify in what region or regions of the world this is focused regarding the types of programs and views pertaining to treatment. 

Line 60 – Should the word “what” be “that”?

Methods: 

Participants – Demographics are provided and inclusion and exclusion criteria are clear.  It is unclear, however, from what area of the world these individuals were found and what methods were used to recruit participants.  Also, where was the study done?  Are all the people together or were there multiple locations involved? 

Procedure – Again, where (what country/region) was the study done?  This is important for knowing the potential applicability regarding statistical analysis. 

Good overview of the types of services provided by the center. 

When were the tests administered to the participants? 

Results:

Here is where the authors mention when the questionnaires were administered (November and May).  This should be in the Methods. 

Discussion:

The discussion goes into detail in ways that support the use of the methods with little discussion as to the limitations of the study.  The authors mention small sample size and limited time, but there also needs to be discussions around the methods, lack of controls, and a need to look at similar programs in different areas/cultures/regions to determine the cultural effectiveness.  In addition, there should also be mention of how this type of study needs to be done in parallel with other studies to compare efficacy of different components.  Also, the authors might consider mentioning the importance of follow-up studies to determine long-term efficacy. 

Conclusion:

The limitations discussed in the conclusion should appear with more detail in the discussion. 

Reviewer 3 Report

The title of the work should be modified, since the word mental health is referred twice

Statistically the work is fine, however I suggest that the authors think about the possibility of making linear regressions between the dimensions studied, as a result of the different questionnaires applied. Or at least indicate why they did not do those tests

The authors should make a greater effort in the conclusions. The conclusions should serve to respond to the objectives of the study with more clarity than the authors indicate.

The aim of the study was to examine (a) the amount of health-related behavior, (b) the level of generalized optimism, (c) the belief about patient’s abilities of coping with difficult situations and obstacles and (d) the subjective sense of social exclusion at baseline and at follow-up among patients with chronic mental issues participating in a psycho-socio rehabilitation program in a community men90 tal health setting. 

Well, the conclusions must serve to respond to each of the objectives formulated

I suggest the authors take these considerations

Reviewer 4 Report

Dear Authors

First of all, I sincerely appreciate that you gave me a chance to review this meaningful research. As a reviewer I have enjoyed reading your work. Below is my recommendation to improve this manuscript. 

  1. Reference and in-text citation: It seems some parts need reference to show what is the background of explanation. For instance, in data analysis (Section 2.3), the authors mentioned "non-parametric test" or "Wilcoxon test." Readers may not know why the authors took certain statistic procedures in this case. The authors do not have to add detail, but it would be great if they add in-text citation behind the statements  so that reader will be able to find the information. 
  2. Result: It would be better if authors use a sign inequality when they describe p-value (e.g. p=0.006 -> p<0.01).
  3. Regarding the methodology, readers may wonder why the authors operated time series analysis. I guess it is because its sample number is too small. It would be great if the authors mention it in the limitation section. 
  4. Also, the limitation of sample size and methodology may lead to additional questions. For instance, the age range is a little bit wide (e.g. 18 - 43) so the participants might be exposed to different social settings that could affect health related behavior and self-efficacy (i.e. I believer the existence of guardianship would make a huge different). So, I recommend the authors mention potential limitation or statement to justify their finding in the conclusion section. 

Yet, I believe that it is a great research idea. I would be happy if I can talk more about the research in the future. Thank you for your hard work!  
